# Broadband and High-Efficiency Multi-Tasking Silicon-Based Geometric-Phase Metasurfaces: A Review

**Jinwei Zeng** [1,2] , **Yajuan Dong** [1,2] , **Jinrun Zhang** [1,2] **and Jian Wang** [1,2,*]

1    Wuhan National Laboratory for Optoelectronics, School of Optical and Electronic Information, Huazhong University of Science and Technology, Wuhan 430074, China

2    Optics Valley Laboratory, Wuhan 430074, China

*    Correspondence: jwang@hust.edu.cn; Tel.: +86-18827006310

**Abstract:** Silicon (Si)-based geometric phase metasurfaces are fantastic state-of-the-art light field manipulators. While the optical metasurfaces generally excel in the micro-control of light with supreme accuracy and flexibility, the geometric phase principle grants them the much-desired broadband phase manipulation property, free from material dispersion. Furthermore, adopting Si as their fundamental material serves as a critical step toward applicable practice. Thanks to the optical lossless feature and CMOS compatibility, Si-based metasurfaces are bestowed with high efficiency and fabrication conveniency. As a result, the Si-based metasurfaces can be perfectly integrated into Si-based optoelectronic chips with on-demand functions, trending to replace the conventional bulky and insufficient macroscopic optical devices. Here we review the origin, physical characteristics, and recent development of Si-based geometric-phase metasurfaces, especially underscoring their important achievements in broadband, high efficiency, and multitasking functionalities. Lastly, we envision their typical potential applications that can be realized in the near future.

**Keywords:** silicon; geometric-phase; multi-tasking; metasurfaces

## 1. Introduction

Metasurfaces are a kind of two-dimensional planar micro-nano optical array, which are composed of artificially customized sub-wavelength unitcells arranged according to a certain rule [1–3]. Metasurfaces can control and manipulate the physical dimensions of light, including phase, polarization, and amplitude, by rationally engineering the response and arrangement of the unitcells [4–7]. After the proposal of their general concept, the metasurfaces have quickly attracted great attention from both academia and industry. The novel electromagnetic wave modulation ability exhibited by metasurfaces means they are widely used in various fields such as anomalous refraction/reflection [2,8], structured light field generation and manipulation [9–15], metalens [16–23], optical holography [24–29], information encryption [30–32], quantum information [33–35], etc.

Early metasurface research frequently used subwavelength plasmonic metal structures as their fundamental element, such as split-ring resonators (SRRs) [36], V-shaped antennas [4], fishnet structures [37], etc., which can enable strong electromagnetic responses with unique optical properties [2]. However, due to the ohmic loss of metal, the inherent absorption losses of plasmonic metasurfaces in the optical frequency range are often non-trivial or even sometimes enormous, therefore their efficiency is physically limited. In recent years, researchers have proposed all-dielectric metasurfaces, which provide a new horizon for efficient metasurface devices. The all-dielectric metasurfaces have a unitcell structural shape that is possibly similar to plasmonic metasurfaces and utilize high refractive index ($n$) dielectric nanostructure to induce strong electrical and magnetic responses.

Compared with plasmonic metasurfaces, dielectric metasurfaces can exhibit very low absorption losses at Near-infrared (NIR) and visible frequencies [38]. Universal dielectric materials such as Silicon (Si) [7,39,40], silicon nitride ($Si_3N_4$) [41,42], titanium dioxide

(TiO$_2$) [20,43,44], gallium nitride (GaN) [21,28], hafnium oxide (HfO$_2$) [45], etc., have been used to design different metasurfaces to achieve efficient optical devices with different functions. For these materials, Si$_3$N$_4$ (*n*~2.0) and TiO$_2$ (*n*~2.4) have been widely used in metasurfaces recently to achieve high efficiency due to their high transparency in the visible range with fabrication convenience, and especially, HfO$_2$ is a favorable material featuring a high-refractive index, especially at an ultraviolet (UV) range [45]. GaN (*n*~2.38) is a III-V material that also has excellent transparency at visible wavelengths with mature fabrication techniques and is especially promising for active metasurface devices [21,28]. The unique merit of Si is its dense dielectric property: It features the highest refractive index (*n*~3.5) among the common dielectric materials, as well as small loss [41,46–49]. Therefore, Si metasurfaces boast a plethora of much-desired properties in size and efficiency, which will be elaborated on in detail in the next sections. In addition, Si also has the much-desired complementary metal-oxide-semiconductor (CMOS) compatibility and low cost, leading to economical mass-production capability.

Except for efficiency, the operational frequency bandwidth is also an essential property of metasurface devices. Since many mainstream metasurfaces are often based on optical resonance, they are usually subjected to very limited bandwidth. To address this bottleneck, geometric phase metasurfaces have been proposed and extensively studied due to their special light manipulation principle [7,8,50–52]. The geometric phase, also named the Pancharatnam–Berry (PB) phase, originates from the interaction of anisotropic unitcells with orthogonally polarized light [53,54]. Geometric-phase metasurfaces arrange unitcells in different in-plane directions, which can control the phase of the orthogonally polarized components of scattered light (relative to the incident light). This phase modulation is highly attractive in integrated photonic devices because it only depends on the rotation angle without material dispersion. In recent years, geometric-phase metasurface devices have been developed and improved with multi-tasking capabilities to further extend their functionalities [26,29,43,55–59].

In this review, we summarize the progress of Si-based geometric phase metasurfaces by reviewing the progress in the past few years and provide some of our perspectives. This article is organized as follows: In Section 2, we provide an overview of all-dielectric Si-based metasurfaces and Mie resonances in dielectric metasurfaces. In Section 3, we introduce the basic principles and applications of the Geometric-phase metasurfaces. In Section 4, we summarize several schemes for multitasking metasurfaces. In Section 5, we provide an overview of tunable optical metasurfaces with controllable functionalities after fabrication. In Section 6, we review the major fabrication techniques and practicalities of metasurfaces. In Section 7 and the final section, we provide a discussion and an outlook on future research directions.

## 2. Dielectric Mie-Type Si-Based Metasurfaces

Si is a widely used dielectric nanomaterial with a high refractive index and compatibility with CMOS technology. The Si-based metasurfaces generally have improved design flexibility and are advantageous in large-scale manufacturing.

One special reason to merit a high refractive index in dielectric metasurfaces is adapting efficient Mie-scattering in the light–matter interaction. Mie-scattering describes the light scattering of a dielectric scatterer with a size similar to or slightly smaller than the incident wavelength, which can grant superior manipulability and flexibility to metasurface devices [38]. Experimental observations of electrical and magnetic multipole Mie resonances at optical frequencies have sparked researchers' interest in high refractive index dielectric materials [60,61]. Earlier studies have found that sufficiently small spherical dielectric nanoparticles can provide significant resonances associated with excitation of the magnetic and electric dipole modes, which can be precisely and analytically described by Mie theory [60]. After that, other dielectric nanoparticle shapes (e.g., cubes, disks, etc.) were investigated to exhibit similar physical properties predicted by Mie-theory. Therefore, electrical and magnetic multipole Mie resonances of high-refractive-index dielectric

nanoparticles at optical frequencies are often used to produce complex electromagnetic responses. Figure 1a exhibits the radiation spectra of fundamental Mie resonances of Si nanodisks, including electric dipole (ED) and magnetic dipole (MD) modes, together with the higher-order multipoles: Electric quadrupole (EQ) and magnetic quadrupole (MQ) modes [38]. Different resonance modes play different roles in the extinction spectra, which suggests that one can modulate the scattering properties of dielectric nanostructures by employing a particular resonance mode. This property has been used to design high-efficiency Huygens' metasurfaces. Huygens' metasurfaces are designed via the equivalence principle by simultaneously tuning the electrical and magnetic dipole modes to implement electrical and magnetic surface impedances that support field discontinuities to achieve very high transmission efficiency and an accumulated $2\pi$ phase change [62–64]. As shown in Figure 1b, Kivshar's group utilized arrays of high-permittivity Si nanodisks simultaneously generating MD and ED resonances, demonstrating 0-to-$2\pi$ phase coverage with a transmission efficiency of more than 55% in the near-infrared regime [63]. In addition, the Si Mie-type metasurfaces can be implemented in efficient functional devices, including a variety of perfect mirrors [46] and magnetic mirrors [65]. As shown in Figure 1c, aided by the electrical and magnetic dipole resonances of the Si-based metasurfaces, large-scale perfect reflectors have achieved an average reflectivity of 99.7% at 1530 nm, exceeding that of metal mirrors [46]. It is amazing that subwavelength-thick metasurfaces composed of Si nanostructured units can achieve close to 100% reflectivity at near-infrared resonance wavelengths [46,47]. In addition, Si-based metasurfaces are also used in optical holography [64,66]. Figure 1d demonstrates a metasurface consisting of sub-wavelength lattices of Si nanopillars, each of which supports several electrical and magnetic Mie resonances. The metasurfaces can produce different types of holograms, experimentally demonstrate a lower diffraction efficiency exceeding 99% at a 1600 nm wavelength, and the transmission efficiency exceeds 90% [66].

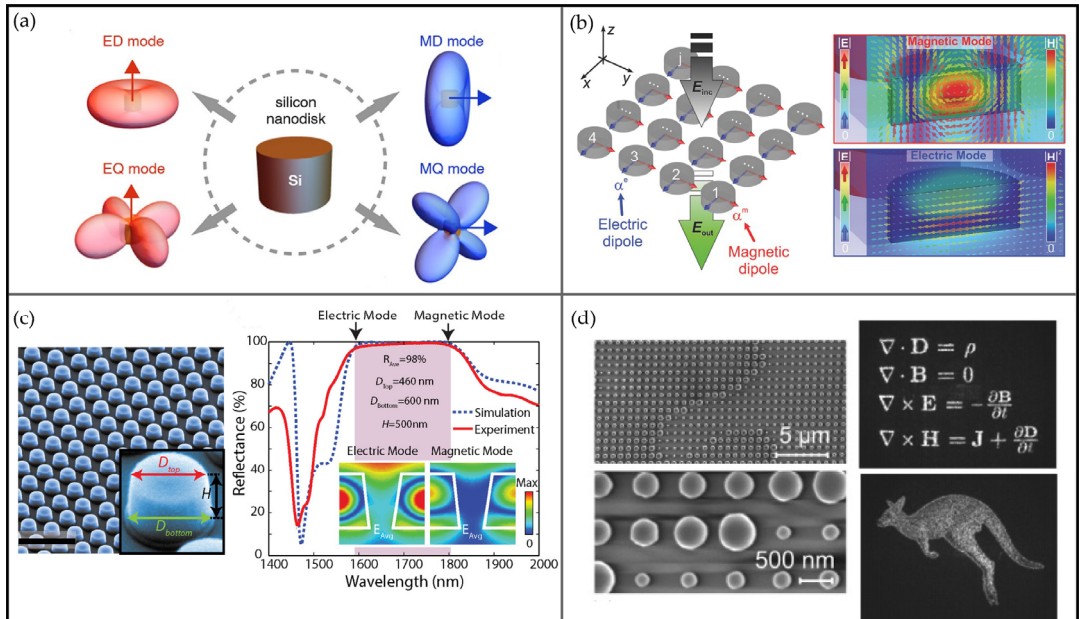

**Figure 1.** (**a**) The radiation spectra of fundamental Mie resonances of Si nanodisks. Reproduced with permission [38]. Copyright 2021, American Physical Society; (**b**) Si nanodisks arrays simultaneously generating MD and ED resonances. Reproduced with permission [63]. Copyright 2015, WILEY–VCH Verlag GmbH & Co. KGaA, Weinheim; (**c**) Si metasurface reflectors have high average reflectivity. Reprinted with permission from [46]. Copyright 2015, American Chemical Society; (**d**) The Si nanopillars metasurface produces different types of holograms. Reprinted from [66]. © The Optical Society.

### 3. Geometric Phase Metasurfaces

Geometric-phase metasurfaces (also named PB phase metasurfaces) are metasurfaces composed of identical artificial microstructures with different in-plane rotation angles. They can realize complex phase modulation of the reflected (or transmitted) wave by changing the rotation angle of the microstructure, thereby realizing the artificial control of the phase gradient or distribution [45]. The development of PB phase metasurfaces is largely based on earlier work on spatially variable subwavelength gratings by Hasman and colleagues, while the basic principle of the PB phase can be revealed by the Jones matrix [50,51]. When the incident light irradiates the anisotropic nanostructure unit along the normal direction, and the nanostructure is rotated at an angle $\theta$ from the *x*-axis, the transmission coefficient of this rotating system can be deduced by the operation of the Jones matrix:

$$\hat{T}(\theta) = \hat{R}(-\theta)\begin{pmatrix} t_o & 0 \\ 0 & t_e \end{pmatrix}\hat{R}(\theta) \tag{1}$$

where $t_o$ and $t_e$ represent its complex transmission coefficients when the polarization of incoming light is along the two axes of the nanostructure, respectively. $\hat{R}(\theta)$ is the rotation matrix, given by

$$\hat{R}(\theta) = \begin{pmatrix} \cos(\theta) & \sin(\theta) \\ -\sin(\theta) & \cos(\theta) \end{pmatrix} \tag{2}$$

Given an incident light with left/right circularly polarized (CP) of $\mathbf{E}_{in}^{L/R}$, the transmitted light $\mathbf{E}_t^{L/R}$ through the anisotropic structure according to Equation (1) can be expressed as

$$\mathbf{E}_t^{L/R} = \hat{T}(\theta)\cdot\mathbf{E}_{in}^{L/R} = \frac{t_o + t_e}{2}\mathbf{E}_{in}^{L/R} + \frac{t_o - t_e}{2}e^{\pm i2\theta}\mathbf{E}_{in}^{L/R} \tag{3}$$

This Equation indicates that when circularly polarized light strikes such a geometric-phase metasurface, the reflected or transmitted wave consists of two components, co-polarized light and cross-polarized light. The component of cross-polarized light obtains phase retardation that is twice the rotation angle of the nanostructure, which gains an additional PB phase of $\pm 2\theta$, where the $\pm$ sign depends on the particular CP state.

Therefore, the design of the geometric phase metasurface is very convenient. By controlling the optical axis direction of the anisotropic unit, the phase of the orthogonal polarization component in the transmitted light can be controlled: If the meta-atom is rotated from 0 to $\pi$, the phase modulation of full $2\pi$ coverage can be achieved. More importantly, it also means that the phase modulation under the geometric phase principle is independent of wavelength and intrinsically dispersion-less.

On the other hand, from Equation (3) it can be observed that although the phase modulation of geometric-phase metasurfaces is independent of wavelength, their energy efficiency, indicated by the polarization conversion efficiency (PCE), can be related to the wavelength. PCE is defined as the energy ratio of the transmitted light, which converts its polarization to the incident beam [39,67,68]. A non-unity PCE means that the output light contains both the modulated component in the orthogonal polarization and the unmodulated component in the original polarization with regard to the incident polarization. The unmodulated component is usually considered noise and may deteriorate the performance of metasurface devices. To maximize the PCE, the geometric-phase elements need to exhibit a perfect half-wave-plate (HWP), which simultaneously has unity transmission and a half-wave phase anisotropy [69]. This requirement is typically challenging since phase anisotropy is usually weak in common materials. Although certain polarized plasmonic resonance in plasmonic metasurfaces could enhance phase anisotropy, the plasmonic metasurfaces usually have relatively low general transmission. As a result, early geometric phase elements based on low index materials or metals are either very thick or have limited efficiency, contributing to a major obstacle for practical geometric-phase metasurface applications.

Si and other similarly dense dielectric materials serve as ideal candidates to address this challenge. First and foremost, Si typically has negligible loss at an optical frequency, so it can support maximum transmission in the form of thin surfaces. Secondly, Si also has a rather high dielectric constant (electric permittivity) among all common dielectric materials, and this feature can be further enhanced with Mie-type resonance. It means Si-based geometric-phase metasurfaces can realize their maximum efficiency while maintaining an optically thin thickness. As a result, they have gained intense favor from the optical community as a promising candidate for nanoscale light manipulators. In 2014, Damin Lin et al. fabricated geometric phase-type metasurfaces using Si nanobeam antennas, which realized the functions of axicons, ultrathin wave plates, blazed gratings, and lenses, as shown in Figure 2a. When an incident beam is a collimated Gaussian left circularly polarized light (LCP) beam, the fabricated axicons create a nondiffracting Bessel beam on the transmission. The metasurface serves as a halfwave plate with π phase retardation between the transverse magnetic (TM) and transverse electric (TE) orthogonal polarizations, and the lens exhibits a focal length of 100 μm with a numerical aperture of 0.43 at a wavelength of 500 nm. When RCP light hits the ultrathin substrate, the transmissive light is converted to an LCP focal spot with a size of 670 nm, which is close to the diffraction limit [7]. In 2016, Alan Zhan et al. designed a lens and vortex beam generator with the SiN-based geometric phase metasurface, shown in Figure 2b. The lens had a high NA ~0.75, achieving approximately 90% transmission efficiency and a focusing efficiency of 40% in visible light. Furthermore, the metasurface shows two distinct orbital angular momentum states [42]. Nonlinear optical wavefront control is also one of the applications of metasurfaces. Metasurfaces composed of high-refractive-index media can provide strong nonlinear responses. Researchers demonstrate the nonlinear wavefront control for the third-harmonic generation (THG) using a Si geometric phase metasurface [58]. As shown in Figure 2c, they encoded phase information and holographic images into the THG of individual Si unitcells using rotational direction and CPL with the PB phase method, and their experiments show polarization-dependent wavefront control and high-fidelity reconstruction of encoded holograms at third-harmonic wavelengths.

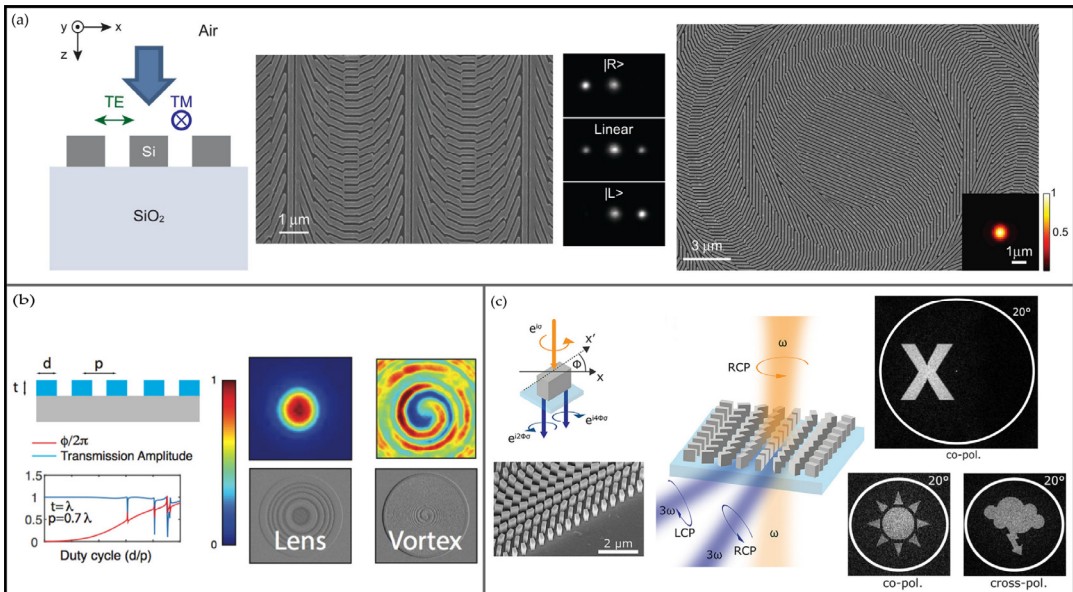

**Figure 2.** (**a**) Si geometric phase-type metasurface realizes the functions of axicons, lenses. Reprinted from [7]. Copyright 2014, The American Association for the Advancement of Science; (**b**) The SiN-based geometric phase metasurface realizes the functions of lenses and vortex beam generator. Reproduced with permission [42]. Copyright 2016, American Chemical Society; (**c**) Si geometric-phase metasurface achieves the nonlinear wavefront control for the third-harmonic generation (THG) and holograms. Reprinted from [58]. Copyright 2019 American Chemical Society.

## 4. Multitasking Metasurfaces

Metasurfaces constitute an excellent integration platform, whose functionality can be readily extensible. It is an important goal of metasurfaces to integrate multiple different functions and tasks into one metasurface platform.

There are two major methods to enable multitasking metasurfaces. The first method is based on the sectioned metasurfaces, in which each section can perform one or a few tasks to form the overall multitasking metasurfaces. One typical example under this principle is interleaved metasurfaces, which divide the complete metasurface into different interleaved regions. All regions are combined according to certain rules to form a complete metasurface device. In 2016, as shown in Figure 3a, using an interleaved Si-gradient metasurface, Dianmin Lin et al. created axial and lateral multifocal geometric-phase metalenses, achieved multiwavelength color imaging with high spatial resolution, and demonstrated that this multifunctional metasurface has simultaneous color separation capabilities of optical imaging [40]. In 2017, Maguid Elhanan et al. designed and fabricated an interleaved geometric-phase metasurface with Si-building blocks by integrating spectropolarimetry and OAM sensing into this metasurface, and according to the schematic diagram shown in Figure 3b, the frequency, polarization, and OAM of light can be controlled and detected simultaneously, increasing the information capacity carried by the metasurface [70].

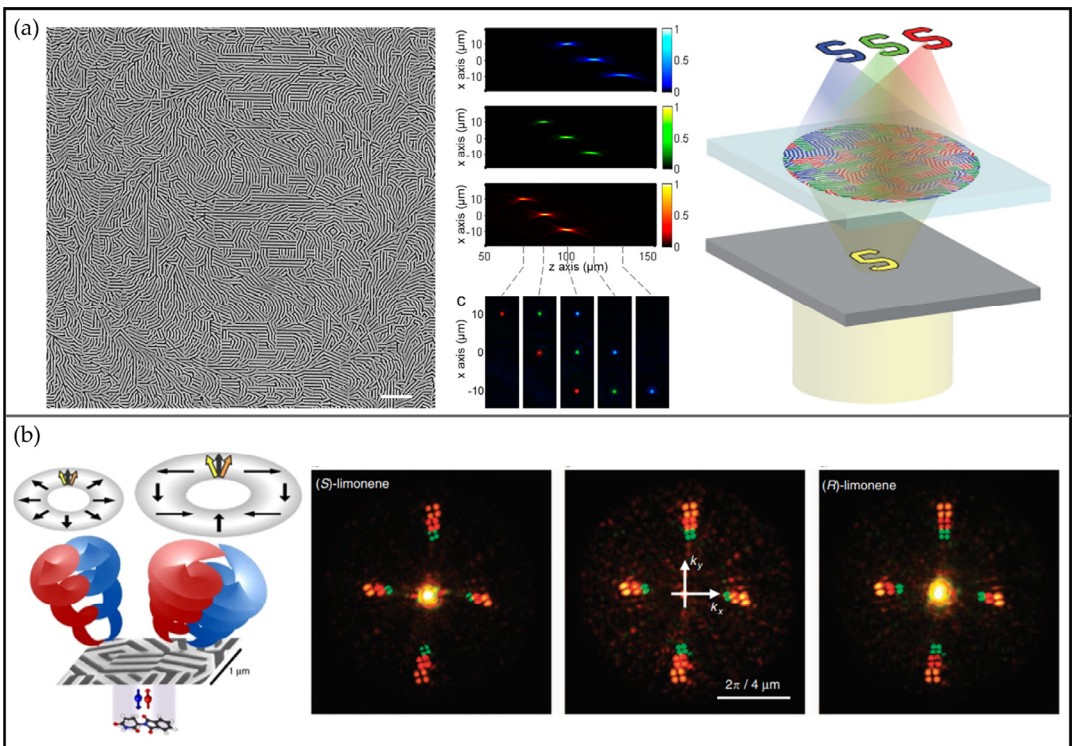

**Figure 3.** The interleaved metasurfaces: (**a**) Interleaved Si-gradient metasurface create axial and lateral multifocal geometric-phase metalenses, achieving multiwavelength color imaging. Reprinted with permission from [40]. Copyright 2016, American Chemical Society. (**b**) The schematic diagram of the Si geometric-phase metasurface integrates spectropolarimetry and OAM sensing, and diffraction patterns of interleaved vectorial vortices generated by using linearly polarized light passing through (S)-limonene, air, and (R)-limonene. Reprinted from [70]. Copyright © 2017, The Author(s).

However, due to the interleaved section, this method also has reduced reconstruction efficiency due to the increased periodicity necessary for interleaving, as discussed in recent literature [71,72]. Besides, the multifunctional meta-devices based on interleaved metasurfaces may be severely affected by functional crosstalk, with a rather large number of interleaved sections. This problem will eventually induce a limited operational efficiency

of approximately 1/N, where N is the number of functions, and potentially restrict their practical application [40]. Rearranging the interleaved meta-atoms could optimize the efficiency, as an important direction to further improve the multi-functionalities of the interleaved metasurfaces.

Another method is multiplexing metasurfaces based on manipulating different physical parameters, which are based on tailored unitcells or multiple physical dimensions of light carrying independent information channels, including wavelength, polarization, phase, linear momentum, orbital angular momentum (OAM), etc., to design the multitasking metasurface devices. Compared to the sectioned metasurfaces, the multiplexing metasurfaces include the multitasking information in each metasurface element, so that they do not need a virtual grouping of the elements. A notable application of the multiplexing metasurfaces is that they can enable simultaneous control of different physical dimensions of light. In 2017, Seyedeh Mahsa Kamali et al. proposed an angle-multiplexed metasurface with a U-shaped cross-section $\alpha$-Si array, which provides independent control of the phase of light at two distinct incident angles. They also demonstrated an angle-multiplexed hologram, which exhibits different holographic images under normal and 30° incident angles with TE polarization [73], and the experimental results are shown in Figure 4a. The Si metasurface has the capability of complete amplitude and phase control for monochromatic grayscale nano-printing and far-field holography. As shown in Figure 4b, Overvig et al. presented a Si metasurface with the capability of complete and independent phase and amplitude control, with multiplexing of the amplitude and phase, enabling holographic encoding in grayscale [74]. In 2020, Hongqiang Zhou et al. proposed a Si nanofins metasurface with OAM multiplexing at different polarization channels, and demonstrated several holographic encryptions [75]. The hologram is shown in Figure 4c. In 2022, Wei Zhu et al. designed a Si polarization multiplexing metasurface, in which each anisotropic unitcell has different phase modulations for orthogonal linearly polarized light beams, as shown in Figure 4d. They also demonstrated multi-channel functionalities, including polarized beam splitting and oppositely charged and differently ordered optical vortexes, which can increase the capacity of the communication channel [76].

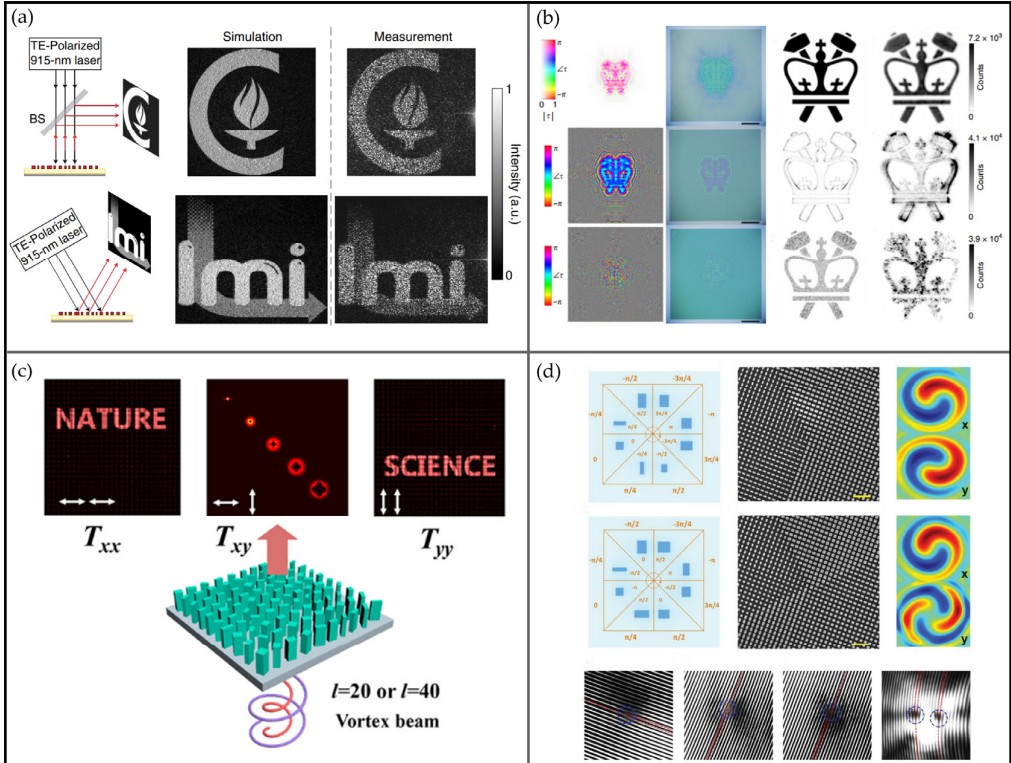

**Figure 4.** (**a**) The angle-multiplexed hologram exhibits different holograms under normal and 30°

incident angles with TE polarization. Reprinted from [73]. CC BY 4.0; (**b**) Si metasurface realizes the multiplexing of amplitude and phase, shows holograms, and compares them with phase modulation only, Reprinted from [74]. CC BY 4.0; (**c**) the OAM multiplexing at different polarization channels Si metasurface demonstrates several holographic encryption. Reprinted from [75]. Copyright 2020 American Chemical Society; (**d**) the schematic diagram of Si polarization multiplexing metasurface achieves a vortex converter, which transforms a Gaussian beam into a vortex beam with helical wavefront and shows experimental results of different OAM and Gaussian beams interference. Reprinted from [76], Copyright 2022 Wiley-VCH GmbH.

It is worth mentioning that, although the above-mentioned metasurfaces can achieve various types of multitasking, but the full dimensional control of light (i.e., the independent, simultaneous, and complete control of phase, polarization, amplitude) has rarely been reported, we believe that designing such full modulation metasurfaces will be an important and promising direction to explore the further potential of efficient and complete light–matter interactions.

## 5. Si-Based Tunable Metasurfaces

Conventional metasurfaces are often not tunable: Their physical properties are fixed after fabrication. Such metasurfaces are usually inactive and unchangeable with time with limited functionalities. Therefore, it is highly desirable to develop tunable metasurfaces for post-fabrication with flexible tunability and functionalities. At present, there has been some related research conducted on tunable metasurfaces [77–84]. Here, we mainly summarize silicon-based tunable metasurfaces and introduce their properties.

The current realization of tunability is mainly based on embedded tunable materials, including liquid crystals (LCs) [85,86], phase-change materials [87,88], etc., or other typical electroactive materials such as two-dimensional materials [82,89,90], III-V materials [78,91] etc. Afterward, the embedded structures can be used to control the properties of metasurfaces and exhibit tunable characteristics via mechanical tuning [80,92,93], electrical tuning [78,82,86,89–91], thermal tuning [94–96], and optical pumping [97–99]. One particular example of mechanical tuning is based on strain. By embedding metasurfaces on flexible material substrates and producing mechanical deformations by bending, stretching, and rolling the metasurfaces, the metasurfaces will discover unique properties that enable novel manipulations of light [80,92,93]. As shown in Figure 5a, a reconfigurable metasurface is composed of Si nanoposts embedded in Polydimethylsiloxane (PDMS), which achieves two imaging modalities including bright-field imaging and 2D differentiation, and demonstrates a NA = 0.25 and a working bandwidth of 60 nm [93]. Second, phase-change materials have been found to be useful for realizing tunable metasurfaces, which have noticeable optical parameter changes between amorphous and crystalline states. Such materials include graphene [81], GeSbTe [87], chalcogenide alloys [88], etc. For example, in Figure 5b, chalcogenide phase-change materials are embedded in silicon nanodisks and placed on an ultra-thin Ge2Sb2Te5 layer to form a reconfigurable metasurface. It enables a dynamic filter and light modulation in the near-infrared regions (O and C telecom bands), with modulation depths up to 70% [87]. Another method exploits the electrically tunable metasurface. By applying external voltage to the structure in which the designed active material substrate is combined with the metasurface array, the structure exhibits tunable performance characteristics under the modulation of the voltage. As shown in Figure 5c, a Mie-type metasurface is composed of Si nanodisks embedded into LCs, applying bias voltage to it, and by controlling the voltage "on" and "off", achieves a large spectral shift of 50 nm and absolute transmission modulation of 75% at the wavelength of 1550 nm. Furthermore, this metasurface has a transmission phase change of up to $\pi$ [86].

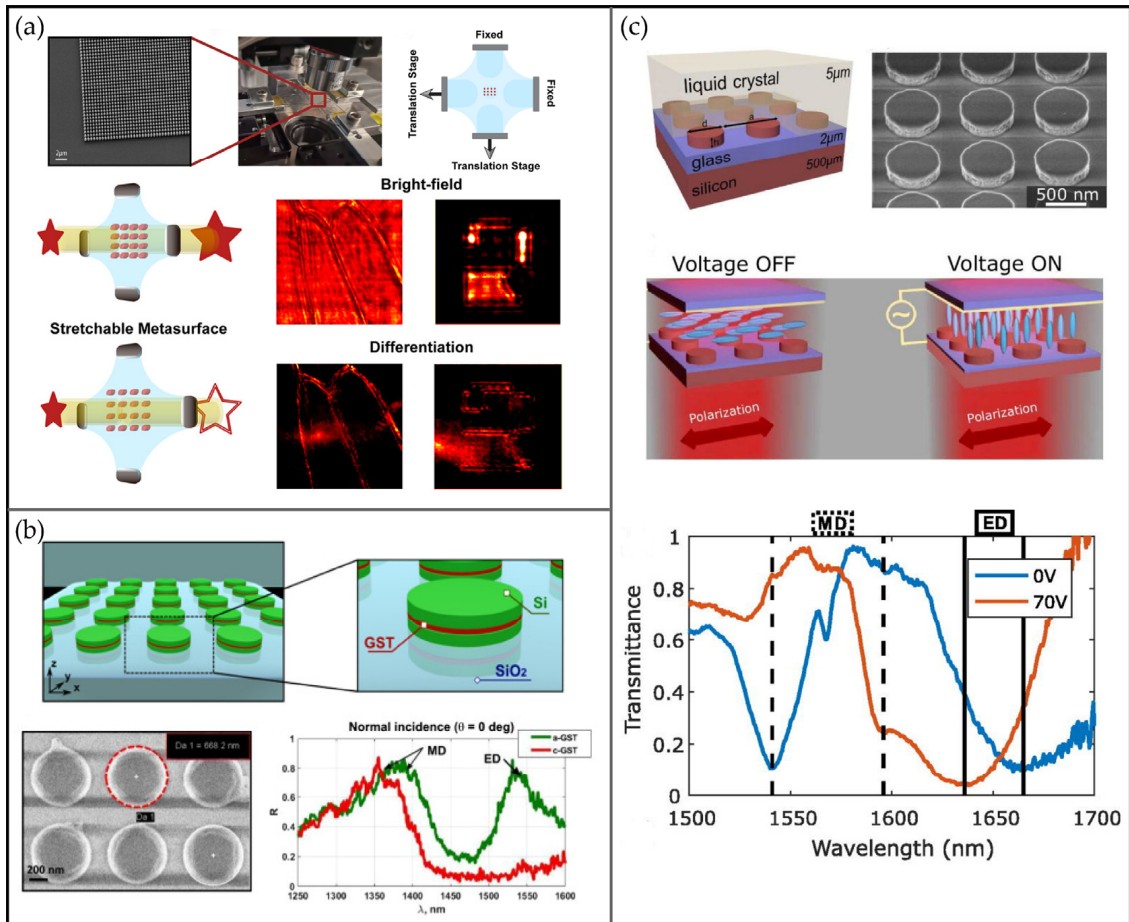

**Figure 5.** (**a**) A reconfigurable metasurface is composed of Si nanoposts embedded in PDMS, which achieves bright-field imaging and 2D differentiation images. Reprinted with permission from [93]. Copyright 2021 American Chemical Society; (**b**) A reconfigurable metasurface consists of chalcogenide phase-change materials embedded in silicon nanodisks. Reprinted from [87]. CC BY 4.0; (**c**) a Mie-resonant metasurface composed of silicon nanodisks embedded into LCs for transmission modulation under electrical bias. Reprinted from [86]. Copyright 2017 AIP Publishing.

Tunable metasurfaces are an important future trend. Indeed, different materials and specially designed metasurfaces combined with different modulation methods enable the tunable control of the optical properties of metasurfaces, including amplitude, phase, polarization, spatial/spectral/temporal responses, etc., and realize different device functions. Combined with Si as the base material, the tunable matasurfaces can be more efficient and economical, which provides a new method for the further development of metasurface-based flat optical elements and devices.

## 6. Fabrication and Practicalities

Metasurface implementation of functional devices depends on nano-machining techniques to construct complex nanostructures. A wide range of nano-manufacturing technologies has been developed and applied, including both top-down and bottom-up approaches [100], to enable efficient and precise fabrication. In general, the top-down approach employs the principles of deposition and lithography to etch a nanostructure. It can process large sizes (from microns to cm magnitude) of material to fit the demand of particular applications. On the other hand, the bottom-up approach constructs a certain nanostructure by controlling the interaction of atoms, molecules, and other nano-objects, employing the evaporation, deposition, or self-assembly of atoms or molecules together. For the fabrication of metasurfaces, the choice of fabrication technique should consider

the inherent advantages and disadvantages of these technologies that satisfy the process requirements, such as the resolution, throughput, reliability, reproducibility, and cost efficiency. In this section, we review various methods and technologies for the fabrication of Si-based metasurfaces.

### 6.1. The Techniques of Top-Down Approach

The top-down approach mainly relies on lithography-after-deposition, which is one of the most matured technologies developed and adapted in metasurface fabrication.

### 6.1.1. Deposition

The top-down approach often starts with deposition. The mainstream deposition methods for metasurface fabrication include but are not limited to physical vapor deposition (PVD) and chemical vapor deposition (CVD). PVD is a technique that uses physical methods to deposit materials. It typically uses high-power electron beams or laser/plasma vaporized target materials, and then deposits the evaporated atoms or molecules on the surface of the substrate to form a thin coating. This process typically does not have a chemical reaction [101,102]. CVD is a process in which vapor phase chemical reactants are introduced into the reaction chamber to react with each other, the reactants undergo a chemical reaction with the substrate to form a non-volatile solid film, and the gaseous by-products of the reaction are pumped out of the reaction chamber [101,102]. CVD includes various processes such as Plasma-Enhanced Chemical Vapor Deposition (PECVD) or Plasma-Assisted Chemical Vapor Deposition (PACVD), Laser-Enhanced Chemical Vapor Deposition (LECVD), etc. In addition, there is a specialized form of CVD, atomic layer deposition (ALD), which is often used to produce extremely thin or highly accurate films of materials. ALD uses a process in which chemical precursors are sequentially introduced into the surface of a substrate, chemically reacted directly with the surface, and deposited layer-by-layer to form a sub-monolayer of thin films, which can deposit thin films precisely at the atomic scale on the substrate surface [103,104]. Different from common CVD, ALD achieves ideal surface flatness and conformal coating by depositing an atomic (molecular) layer per cycle, and the film thickness is precisely controlled by the number of ALD cycles [103,104]. While PVD and CVD have been the most popular deposition methods, ALD has been increasingly attractive in metasurface fabrication recently. Because it exhibits large-area periodicity, uniformity, and conformality, it is advantageous for fabricating complex three-dimensional (3D) nanostructures. In addition, compared with PVD and other methods, the damage to the sample is small and the pollution is low [105].

### 6.1.2. Mask-Based Lithography Method

Mask-based lithography is an indirect lithography method. It typically uses a mask in which a certain beam can write nanoscale patterns. Later, a series of physical and/or chemical processes are used to make the patterns on the sample and remove the mask. The mainstream mask-based lithography methods for metasurface fabrication are introduced as follows.

Firstly, photolithography is a technique of transferring the patterns from photomasks to the substrate by a photoresist, and then combining the etching, depositing, and lift-off process, while target nanostructures are formed on the substrates, including mask preparation, pattern formation and metastasis (glue, exposure, development), film deposition, and etching. It is a mature fabrication technology suitable for the precise mass production of large-area metasurfaces. For example, researchers fabricated a metasurface-based half-wave plate using deep ultraviolet (DUV) photolithography on a 12-inch Si wafer, and the polarization conversion efficiency of HWP was as high as $95.6\% \pm 0.8\%$ [106]. Researchers also fabricated a metalens working at a wavelength of 940 nm with a numerical aperture (NA) of 0.496 on a 12-inch glass wafer through 193-nm ArF DUV immersion lithography combined with a pattern transfer process [107]. However, due to the small size of metasurface structures, the minimum achievable structure size of photolithography is limited by

optical diffraction. Furthermore, the material base of photolithography is subjected to the chemical/mechanical process.

Secondly, electron-beam lithography (EBL) is a basic nano-machining technique, which in principle works to change the solubility during the development process by highly focused electron beams, combines the graphic transfer process, and then makes the designed structure. It has a high resolution, and is currently widely used in metasurface fabrication. The general process is shown in Figure 6a. In particular, the processing of Si-based metasurfaces by EBL has been carefully established: For example, in 2014, Brongersam et al. used EBL-processed Si nanoantennas to realize wave plate gratings, lenses, and axicons [7]. In 2015, the Si-based metasurface designed and processed by Arbabi et al. achieved complete control of phase and polarization, and the schematic diagram and SEM are shown in Figure 6b [39]. Large-scale metalenses were also realized using EBL-fabricated Si geometric-phase metasurfaces [108,109], and Figure 6c demonstrates a metalens with NA = 0.98 and a bandwidth of 274 nm, with a focusing efficiency of 67% at a 532 nm wavelength [109]. In particular, EBL can be combined with ALD to process precise and complex multi-layer structures, as shown in Figure 6d, as Katsuya Tanaka's group fabricated a chiral bilayer Si metasurface, which achieved high circular dichroism of 0.7 and optical activity of $2.67 \times 105$ degree/mm. The high-quality conformality of ALD enables multi-layer complex processing to achieve precise chirality control [77]. At present, Si-based metasurfaces based on EBL processing have been widely used in holography, quantum optics, multitasking metasurfaces, etc. The drawback of the EBL process is that it is typically slower than photolithography, and the material bases are also subjected to the chemical/mechanical etching process.

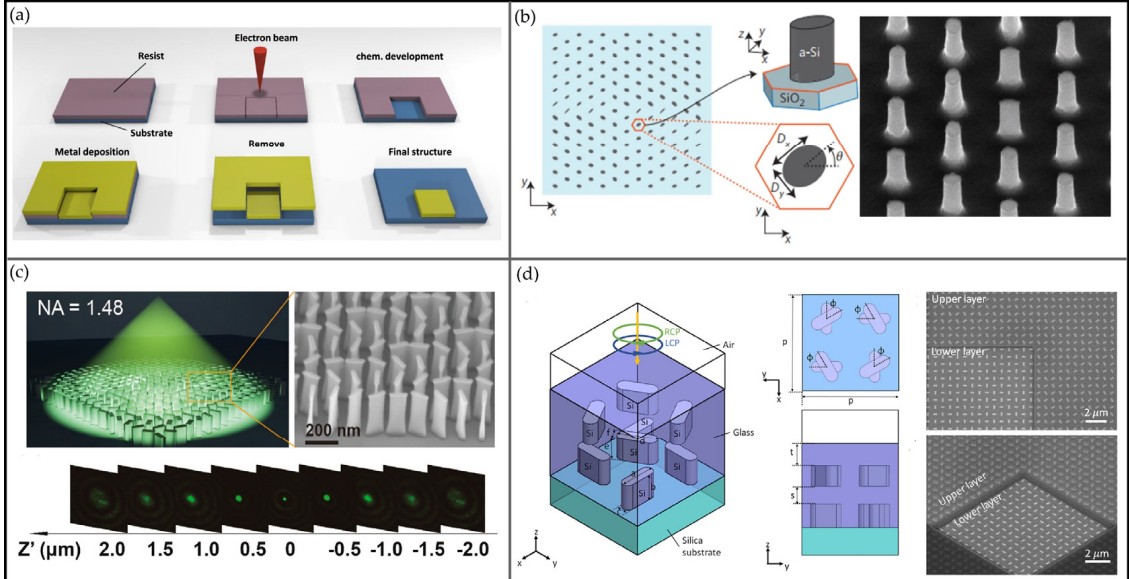

**Figure 6.** (**a**) The general process of the EBL approach, reprinted from [110]. Copyright 2022 IOP Publishing Ltd.; (**b**) the Si metasurface achieved complete control of phase and polarization. Reprinted from [39]. Copyright 2015, Nature Publishing Group; (**c**) the SEM of EBL-fabricated large-scale Si geometric phase metalens, and the imaging results of the lens at different positions. Reprinted with permission from [109]. Copyright 2018 American Chemical Society; (**d**) the schematic diagram and SEM image of a chiral bilayer Si metasurfaces. Reprinted with permission from [111]. Copyright 2020 American Chemical Society.

### 6.1.3. Direct-Writing Lithography Method

Direct-writing lithography has been developed for the processing of various materials. These technologies are mask-less, and therefore convenient to prepare, but typically much slower than the mask-based method.

One well-established direct-writing method is focused-ion-beam (FIB) lithography. This method has a one-step etching process, removing the atoms from the surface or depositing the atoms onto the surface of the sample by bombarding the surface with an ion beam (usually $Ga^+$ ions). During processing, it is mask-free and there is no need for a graphical transfer process. Through a carefully developed process recipe, the FIB can fabricate a rather complex nanostructure for a wide range of base materials. Based on the features of the FIB fabrication method, it has a plethora of applications in small-area manufacturing of metasurfaces. For example, Maxim V. Gorkunov et al. designed and fabricated a chiral visible-light Si metasurface by FIB, which combines high transparency with strong optical chirality, the schematic diagram and SEM image of the metasurface are shown in Figure 7a–d [112]. As shown in Figure 7e, Kseniia V. Baryshnikov's group fabricates the Si parallelepiped metasurface, which allows excitation of the magnetic dipole moment, leading to broadening and enhancement of absorption, and the resonant electric quadrupole moment leads to the enhancement of reflection [113]. The downside of FIB includes the often-deteriorating beam damage, high cost, and limited total fabrication scale. Therefore, the FIB method is mostly employed in early proof-of-principle research on metasurfaces, rather than in the mass-production period.

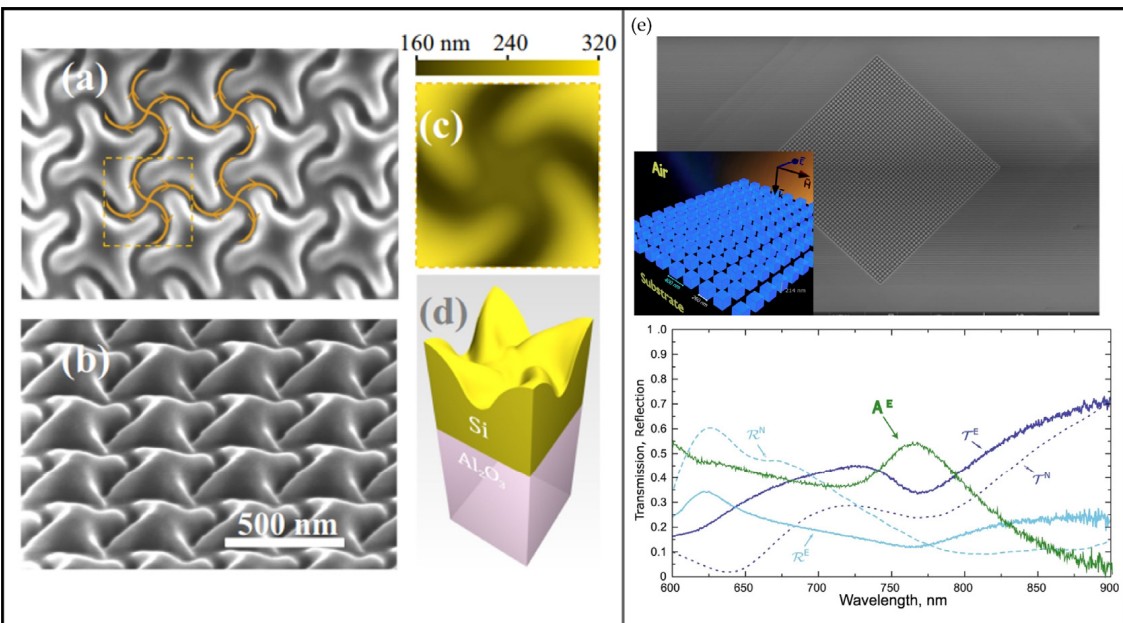

**Figure 7.** (**a–d**) The schematic diagram and SEM image of chiral visible light Si metasurface by FIB. Reprinted from [112]. CC BY 4.0; (**e**) the SEM of Si parallelepiped metasurface, and the transmission and reflection spectra of the metasurface. Reprinted from [113]. CC BY 4.0.

### 6.1.4. Other Methods

Moreover, laser direct writing technology has also been developed to fabricate large-area metasurfaces, including direct-laser-write lithography (DLWL) and laser interference lithography (LIL). DLWL can project patterns directly on the photoresist via a software-defined pattern projected on a computer, controlled with and on the optical system [114,115]. LIL exploits the interference between two or more coherent light waves to yield large-area metasurfaces [116]. They can reduce costs, but they are limited to periodic patterns.

Nanoimprint lithography (NIL) technology achieves pattern transfer by transferring the mechanical deformation of the medium rather than changing its chemical properties [117,118]. The basic principle of nanoimprint technology is to transfer the pattern on the mask to the substrate through a transferred medium that mostly uses polymer films, such as Polymethylmethacrylate (PMMA), PDMS, etc. [117]. This technology has the advantages of large-area processing and low cost, but it still requires high-resolution

equipment to manufacture the mask, and other etching techniques need to be used to remove the excess glue after imprinting.

### 6.2. The Techniques of Bottom-Up Approach

In addition, the bottom-up fabrication methods can also be applied in large-area fabrications, one of the most popular techniques is Self-assembly.

This method uses the polystyrene sphere array formed by colloidal self-assembly as a mask, and then combined with the etching or deposition process, it can be used for the processing and manufacturing of complex metasurfaces with large sizes. For example, based on self-assembly, Jason Valentine's group fabricated a large-scale perfect reflector made of Si cylindrical resonators with an average reflectivity of 99.7% at 1530 nm, exceeding that of metal mirrors [46]. This technology has the advantages of simplicity and low cost but can only fabricate simple patterns arranged periodically.

Notably, bottom-up and top-down approaches may not be strictly exclusive, but their hybrid has been developed to gain the advantages of both sides. For example, Ji Zhou's group designed a flexible Si metasurface via the nanosphere lithography (NSL) technique, which is a combination of the bottom-up approach and top-down approach, shown in Figure 8a. The fabricated Si cylinder metasurface implements a Mie resonance-based sensor, which can detect the applied strain and surface dielectric environment by measuring transmission spectra, as shown in Figure 8c [119].

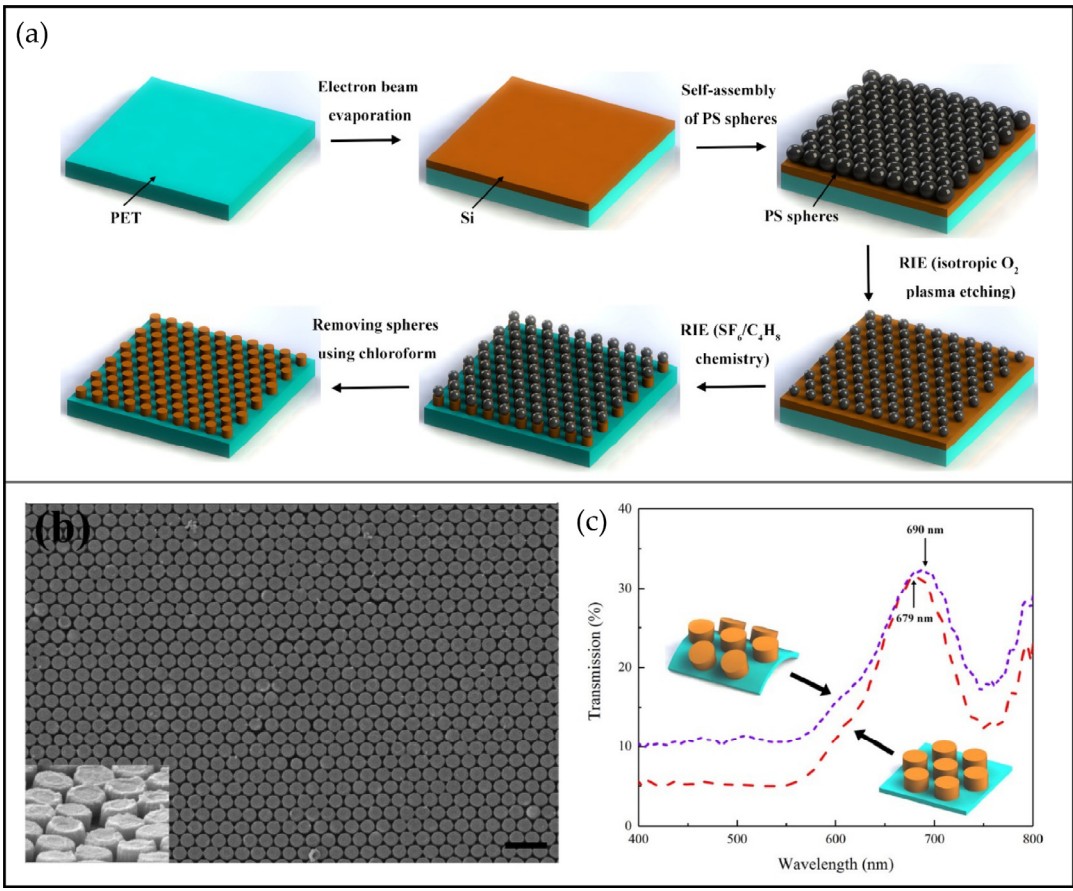

**Figure 8.** (**a**) The process of designing a flexible Si metasurface with a bottom-up approach and top-down approach; (**b**) The SEM of the flexible Si metasurface; (**c**) The transmission spectra of the metasurfaces with and without strain. Reprinted with permission from [119]. © The Optical Society.

In the near future, the development trend of metasurfaces will always be lower cost, larger area, and more complex structures. To achieve mass production from the laboratory to a business model, mature processing technology is required. According to the properties

and challenges of various fabrication technologies, a combination of various technologies can be developed, improved, and combined to fit these goals.

## 7. Discussion

Based on the above discussions, here we summarize several typical features of metasurfaces that can be employed in the Si metasurfaces, introduce their main properties, and review some representative applications, listed in Table 1.

**Table 1.** Guidelines for several applications of different types of metasurfaces.

| Metasurfaces Types | Properties | | Applications | |
|---|---|---|---|---|
| Dielectric Mie-type Si-based metasurfaces [46,47,62–66] | (1) | High transmittance Efficiency; | (1) | Perfect mirrors and magnetic mirrors; |
| | (2) | Low absorption loss | (2) | Metahologram |
| Geometric phase Metasurfaces [7,42,56,58] | (1) (2) | Broadband; Large fabrication tolerance | (1) (2) (3) | Metalens; Metahologram; Vortex beam converter |
| Multitasking Metasurfaces [40,70–75] | (1) (2) | Multi-dimensional modulation; Multiple different functions | (1) (2) (3) | Metahologram; Structure light generator; Metalens |
| Si tunable metasurface [77,83,84,92,96] | (1) | Tunable control of the optical properties | (1) (2) (3) | Spatial light Modulation; Metahologram; Optical switching |

Currently, a wide range of fabrication techniques has been developed for metasurface fabrication. We discuss the mainstream fabrication methods in Section 6, and their main properties and typical applications are listed in Table 2. Photolithography is a mature fabrication method, it transfers the pattern on the mask to the substrate with the help of photoresist, including exposure, development, deposition, lift-off, etching, multiprocess fabrication, etc. Recently, it has also been used for the large-scale fabrication of metasurface devices with various functionalities, and it has fast fabrication and is suitable for mass production [106,107,120–122]. EBL and FIB are processed by a point-by-point scanning method. EBL requires multiple processes, while FIB is a one-step etching process. They are relatively slow to process and require a great deal of processing time for the preparation of large-scale nanostructures, so they are suitable for small-area fabrication. DLWL uses laser beams with variable intensity to directly scan and process materials, and it can draw arbitrary 3D nanostructures [114,115]. The processing efficiency of DLWL is relatively higher than that of EBL and FIB, and can be used for large-area fabrication [123]. NIL pattern nanostructures by creating mechanical deformation or transferring the pattern on the mask to the substrate, it can be fast replication. If the template is relatively large, NIL can be used for large-area manufacturing and mass production [117,118]. Self-assembly lithography also can be used for large-area fabrication, as it uses colloidal self-assembly to form an array as a mask, and then combined etching or deposition multiprocess fabrication, which has the advantages of simplicity and low cost, but can only process simple patterns of periodic arrays [46,119]. These methods will offer great potential for future metasurfaces in terms of versatility and functionality.

Indeed, practical metasurface devices demand the design and processing of macroscopic devices with large-scale, complex nanostructures that are compatible with low loss materials. As a result, they also demand the improvement of various processing and manufacturing technologies. These methods aim to realize the design, processing, and mass

production of various complex nano-structures to make multi-functional metasurfaces, which are expected to produce new compact optical devices and various small portable instruments, such as novel cameras for mobile phones, wearable displays for augmented reality, etc.

**Table 2.** The properties of different fabrication methods of metasurfaces.

| Fabrication Approach | Properties | | Applications | |
|---|---|---|---|---|
| Photolithography [106,107,120–122] | (1)<br>(2)<br>(3)<br>(4) | Mask based multiprocess fabrication;<br>High resolution;<br>Large-area fabrication;<br>Fast fabrication and Mass producible; | (1)<br>(2) | Wave plate;<br>Metalens |
| Electron-beam lithography (EBL) [7,39,77,108,109] | (1)<br>(2)<br>(3)<br>(4) | Mask based multiprocess fabrication;<br>High resolution;<br>Small-area fabrication;<br>Relative slow fabrication | (1)<br>(2)<br>(3) | Wave plate;<br>Metalens;<br>Metahologram |
| Focused-ion-beam lithography (FIB) [112,113] | (1)<br>(2)<br>(3)<br>(4)<br>(5) | Direct writing;<br>High resolution;<br>Small-area fabrication<br>Broader material base;<br>Relative slow fabrication | (1)<br>(2) | Metahologram;<br>Optical vortex generation |
| Direct-laser-write lithography (DLWL) [114,115] | (1)<br>(2)<br>(3)<br>(4) | Direct writing;<br>Low resolution;<br>Large-area fabrication;<br>3D fabrication | (1)<br>(2) | Optical Vortex Modulator;<br>Optical absorber |
| Nanoimprint lithography (NIL) [117,118] | (1)<br>(2)<br>(3)<br>(4) | Mechanical transfer multiprocess fabrication;<br>High resolution<br>Large-area fabrication;<br>Fast replication and Mass producible | (1)<br>(2) | Metalens;<br>Optical Sensor |
| Self-assembly lithography [46,119] | (1)<br>(2)<br>(3)<br>(4) | Template transfer multiprocess fabrication;<br>Low resolution;<br>Large-area fabrication;<br>Relative slow fabrication | (1) | Perfect reflector |

## 8. Conclusions

In summary, the rapid development and special optical properties of metasurfaces have made them an important platform for optical devices. In this review, we discuss the origin and physical properties of Si-based geometric metasurfaces, summarize several multitasking metasurfaces, and review the current mainstream metasurface fabrication methods. Through the discovery and utilization of new structures and patterns, together with the selection of efficient, accurate, and low-cost fabrication technologies, Si-based geometric phase metasurfaces with typical and special characteristics contribute to novel and promising optical devices that can reshape our daily lives.

**Author Contributions:** This manuscript was prepared and written with the collaboration of all authors. All authors have read and agreed to the published version of the manuscript.

**Funding:** This research was funded by the National Natural Science Foundation of China (NSFC) (62125503), the Key R&D Program of Hubei Province of China (2020BAB001, 2020BAA007, 2021BAA024), the Key R&D Program of Guangdong Province (2018B030325002), and the Science and Technology Innovation Commission of Shenzhen (JCYJ20200109114018750).

**Institutional Review Board Statement:** Not applicable.

**Informed Consent Statement:** Not applicable.

**Data Availability Statement:** Not applicable.

**Acknowledgments:** The authors acknowledge Jin Zheng from Aname for the helpful discussion about ALD.

**Conflicts of Interest:** The authors declare no conflict of interest.

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
