# Peer review of "Broadband and High-Efficiency Multi-Tasking Silicon-Based Geometric-Phase Metasurfaces: A Review"

_photonics, doi:10.3390/photonics9090606_

Round 1
Reviewer 1 Report
The manuscript is too general and lacks a real work of synthesis and of a real review and therefore I feel that it is not fit for publication.
Author Response
We appreciate this comment from the reviewer! And we make a substantial revision to improve our manuscript to show more specific and detailed, we hope the current manuscript shows complete and important to the field, for the reviewer’s convenience, the detail revision marks provide in the revised manuscript .
Reviewer 2 Report
The paper by Zeng et al is an extensive review on silicon metasurfaces. The topic is very interesting and timely for the photonics community The authors present a quite comprehensive overview on this topic that, in my opinion, deserves publication.Nevertheless, I believe that the paper could be improved by addressing the following points
Section 2 could benefit from a short explanation about the working principle behind Huygens metasurfaces
The sentence corresponding to the lines 151-160 should be extended, rephrased and made clearer
The authors describe several applications, but they should ​also comment on what are the most interesting scenario or applications in which metasurfaces could provide a significant breakthrough
The insertion of tables comparing device performances and applications could help the reader
Tunable metarsurfaces could deserve a dedicated section as they represent a very promising toolAuthor Response
Please see the attachment.

Reviewer 3 Report
This paper gives a review of silicon-based geometric-phase metasurfaces, contains its advantages compared with plasmonic metasurfaces, its applications in Mie resonance, geometric-phase metasurfaces and multitasking metasurfaces, as well as its fabrication techniques. However, several similar reviews have been published recently, like application areas reviewed in J. Phys. D: Appl. Phys. 54 (2021) 383002 and fabrication methods reviewed in J. Opt. 24 (2022) 033001, etc. Therefore, the analysis of the concepts should be more in-depth to show the reader something more valuable.
1. The paper mentioned lots of dielectric materials used to form the metasurfaces, like Si, SiN, TiO2, GaN, HfO2. Although the authors focused on Si-based applications, the pros and cons of those materials should be discussed rather than just CMOS compatibility.
2. The three applications introduced in the paper are not clear. A table with main specifications and contributions from different research groups will make the discussion much clearer.
3. Also for the fabrication method, I think it is not easy to fabricate such devices. However, the authors just showed some well known nanofabrication methods. After reading, I do not have an idea of how to fabricate them successfully and which method should I choose when I want to fabricate a specific device?
4. For the outlook, the authors mentioned tunability, but what kind of enhancement or magic will happen in the applications discussed in the paper with tunability? As shown in the review paper 'Rep. Prog. Phys. 85 (2022) 036101', there are already lots of results in this field, is there any other trend of silicon-based metasurfaces?
Round 2
Reviewer 3 Report
The authors give clear answers to the questions and carefully revise the paper. I agree that this version can be accepted for publication.